# Characterization of the Antibiotic and Copper Resistance of Emergent Species of Onion-Pathogenic *Burkholderia* Through Genome Sequence Analysis and High-Throughput Sequencing of Differentially Enriched Random Transposon Mutants

**DOI:** 10.3390/pathogens14030226

**Published:** 2025-02-25

**Authors:** Jonas J. Padilla, Marco A. S. da Gama, Inderjit Barphagha, Jong Hyun Ham

**Affiliations:** 1Department of Plant Pathology and Crop Physiology, Louisiana State University Agricultural Center, Baton Rouge, LA 70803, USA; jjpadilla@agcenter.lsu.edu (J.J.P.); ibarphagha@agcenter.lsu.edu (I.B.); 2Department of Agronomy, Universidade Federal Rural de Pernambuco, Recife 52171-900, PE, Brazil; marco.gama@ufrpe.br

**Keywords:** antimicrobial resistance, copper resistance, transposon-insertion sequencing, high-throughput identification of random mutants

## Abstract

The prevalence of antimicrobial resistance (AMR) in bacterial pathogens resulting from the intensive usage of antibiotics and antibiotic compounds is acknowledged as a significant global concern that impacts both human and animal health. In this study, we sequenced and analyzed the genomes of two emergent onion-pathogenic species of *Burkholderia*, *B*. *cenocepacia* CCRMBC56 and *B*. *orbicola* CCRMBC23, focusing on genes that are potentially associated with their high level of antibiotic and copper resistance. We also identified genes contributing to the copper resistance of *B*. *cenocepacia* CCRMBC56 through high-throughput analysis of mutated genes in random transposon mutant populations that were differentially enriched in a copper-containing medium. The results indicated that genes involved in DNA integration, recombination, and cation transport are important for the survival of *B. cenocepacia* CCRMBC56 in copper-stressed conditions. Furthermore, the fitness effect analysis identified additional genes crucial for copper resistance, which are involved in functions associated with the oxidative stress response, the ABC transporter complex, and the cell outer membrane. In the same analysis, genes related to penicillin binding, the TCA cycle, and FAD binding were found to hinder bacterial adaptation to copper toxicity. This study provides potential targets for reducing the copper resistance of *B. cenocepacia* and other copper-resistant bacterial pathogens.

## 1. Introduction

The advent of antibiotics has significantly reduced post-surgical infections, leading to a marked decline in the mortality rates associated with bacterial infections [1,2]. Additionally, the development of antibiotics has extended the human lifespan throughout the 20th century [3]. Nevertheless, the intensive usage of antibiotics creates substantial selection pressure, facilitating the survival and reproduction of antibiotic-resistant bacteria, thereby significantly contributing to the spreading of antibiotic-resistant microorganisms [4,5,6].

Antimicrobial resistance (AMR) occurs when pathogenic microbes undergo genetic changes that confer resistance to the antibiotic treatments designed to eliminate them. This phenomenon is often termed a “silent pandemic” due to its gradual yet potentially catastrophic impact on global health [7,8]. The incidence of AMR in agriculture represents a significant global issue, impacting both animal and human health [9], and has been increasing since 2001, with a recorded overall incidence of 28% and 18% for the conventional and organic farms, respectively [10]. The link between AMR in livestock farming and crop cultivation is intricate and interconnected. Animal waste used as fertilizer may introduce AMR bacteria into agricultural soils. This process may enrich the presence of antibiotic resistance genes (ARGs) within the fertilized soil and facilitate the transfer of ARGs from animal manure to plant root and leaf endophytes through horizontal gene transfer (HGT) [11,12]. The current research results and existing knowledge indicate a significant disparity in the volume of research conducted on antimicrobial resistance (AMR) in animal husbandry versus crop agriculture. This contrast highlights the necessity of enhancing AMR research initiatives in crop farming to achieve a more comprehensive understanding of AMR in agriculture overall. Addressing this research gap is essential for developing effective and coordinated strategies to combat AMR in all agricultural sectors.

For nearly 140 years, copper-based antimicrobial compounds (CBACs) have been extensively employed in agriculture to manage various plant diseases [13], including fire blight in pome [14], walnut blight [15], and stone fruit canker [16]. CBACs are used as antimicrobial agents that serve as protective barriers on plant surfaces [13]. They are accessible in various forms, including inorganic and organic copper, as well as copper nanoparticles. In 1956, copper sulfate, a soluble inorganic salt, was introduced in the United States as a general crop protective compound [17]. An additional example is Kocide 3000-O (Certis USA, Columbia, MD, USA), an organic copper pesticide that has been approved by the Environmental Protection Agency (EPA) and is typically used in producing organic crops. These CBACs are responsible for the release of copper ions, which disrupt the membranes of bacterial cells [18,19], interfere with energy transport [20], inactivate certain enzymes [19], and degrade nucleic acids [21]. The utilization of copper in agricultural cultivation offers broad-spectrum and long-lasting protection, while simultaneously facilitating the restoration of wounds in plants [13,18]. Although there are concerns regarding resistance development, research has demonstrated that numerous clinical microorganisms possess modest copper-resistance levels [21]. This renders copper an appealing alternative for the treatment of antibiotic-resistant pathogens.

*Burkholderia cenocepacia* and *B. orbicola*, previously classified as lineages IIIA and IIIB of *B. cepacia* [22,23], belong to the *B. cepacia* complex (Bcc), which includes at least 28 species that show a strong phylogenomic relationship [22,24]. Even though *B. cepacia* (syn. *Pseudomonas cepacia*) has long been recognized as the causal agent of onion sour skin [25], strains of *B. cenocepacia* and *B. orbicola* have been frequently isolated from onions with sour skin symptoms over the past years [26,27]. These bacterial species are motile, rod-shaped, and Gram-negative, with a genome size of over seven megabases [22,28], which aids in their ability to adapt to their host and their microbiological versatility [22,29]. These pathogens possess various virulence factors, including those involved in biofilm formation, protease activity, and siderophore production, which facilitate their invasion of hosts, allowing for their growth and multiplication that lead to abnormal physiological and morphological changes in the hosts [30,31]. These bacterial species not only infect onion plants but also attack and cause rot in other plant species [32,33]. Aside from being pathogenic to multiple plant species, *B. cenocepacia* and *B. orbicola* can be an opportunistic human pathogen causing severe health complications in immunocompromised individuals, especially those with cystic fibrosis (CF) [34]. Although the majority of the strains isolated from the clinic context are *B. cenocepacia* [22,23], infection with these bacteria in CF patients can result in the potentially fatal “*cepacia* syndrome”, which may culminate in lung failure and an increased mortality risk [22,35,36]. These bacteria possess extensive antibiotic resistance, which complicates the treatment of infections [36,37]. They exhibit resistance to nearly all antibiotics currently available for clinical use, such as aminoglycosides, quinolones, and β-lactams [36]. Thus, the resistance of these bacteria to various antimicrobial agents has become a major focus of scientists in the medical field, as it relates to treating the diseases caused these pathogens.

This report provides the complete genome assemblies of two emergent onion pathogens, *B. cenocepacia* CCRMBC56 and *B. orbicola* CCRMBC23, which can serve as reference genomes for onion-associated microorganisms belonging to the Bcc. Through comparative genomic analyses, we demonstrated the similarities of ARGs present in *Burkholderia* species isolated from humans, plants, and the environment. Lastly, we predicted the genes and gene modules that are important for the survival of *B. cenocepacia* under copper-stressed conditions by conducting transposon-insertion sequencing.

## 2. Materials and Methods

### 2.1. Bacterial Strains and Culture Media

The two onion-pathogenic *Burkholderia* strains, *B. cenocepacia* CCRMBC56 and *B. orbicola* CCRMBC23, were originally isolated from onion fields in Pernambuco State, Brazil [27]. Bacterial cultures were first grown from a glycerol stock culture on Luria–Bertani (LB) agar plates. The overnight-cultured bacterial cells were then resuspended in LB broth with the concentration adjusted to achieve an OD_600_ of 0.1. A 50 µL aliquot from this bacterial suspension was inoculated in 50 mL of LB broth and incubated in a shaking incubator at 190 rpm at 28 °C overnight.

### 2.2. DNA Extraction, Whole Genome Sequencing, and De Novo Genome Assembly

A 3 mL aliquot from an overnight culture of *B. cenocepacia* CCRMBC56 and *B. orbicola* CCRMBC56 was used for DNA extraction. The bacterial genomic DNA was isolated using the GenElute Bacterial Genomic DNA kit (Sigma-Aldrich, Inc., St. Louis, MO, USA), following the protocol provided. The DNA samples were sent to the DNA Sequencing Center at Brigham Young University (Provo, UT, USA) for genome sequencing. Sequencing was performed using the PacBio Sequel II SMRT Cell instrument, producing circular consensus reads of 80–100 Kb. The generated hi-fi reads were de novo assembled using the Flye (v 2.9.3-b1797) software [38] employing three rounds of polishing iterations. The redundant sequences and the misassembled regions were fixed, and the assembled genomes were circularized using the clean and the fixstart functions embedded in the Circlator (v 1.5.5) software [39]. The assembled genomes were evaluated using the BUSCO (v 5.7.1) and the QUAST (v 5.2.0) tools [40,41].

### 2.3. Phylogenomics and ARGs Prediction

To estimate the relatedness of our onion-pathogenic *Burkholderia* strains to the other *B. cenocepacia lato sensu* (i.e., strains from lineages IIIA and IIID) isolates, 27 publicly available completely assembled genomes of *B. cenocepacia* were downloaded from the NCBI database, and the genome of *B. glumae* strain BGR1 was used as an outgroup for the analysis. The average nucleotide identity (ANI) of each strain to each other was computed by aligning the genome sequences using the BLAST algorithm in the JSpeciesWS tool [42]. The resulting ANI values were used to perform hierarchical clustering and create a dendrogram. The de novo assembled genomes and the downloaded assemblies were annotated using the Bakta tool (v 1.9.2) [43]. The generated protein sequence file for each genome was used as the input file for comparative genomics using Orthofinder (v 2.5.5) [44] to create a phylogenetic tree based on the gene trees generated from orthogroups across all strains. To characterize the antimicrobial resistance genes (ARGs) present in each genome, the nucleic acid and protein fasta files, as well as the annotation file in generic feature format (gff3), of each genome were used as input files in the NCBI-AMRFinderPlus pipeline [45]. The presence of different ARGs and their homologies to known ARGs in the NCBI databases were compared across all the genomes selected for this study (Figure 1A).

### 2.4. Transposon Mutagenesis and Copper Resistance Screening

*B. cenocepacia* CCRMBC56 was utilized to create a transposome library by conjugating it with *E. coli* S17-1λpir (pUT::miniTn*5*Cm), an *E. coli* strain carrying miniTn*5*Cm, a Tn*5* transposon derivative with a chloramphenicol resistance marker. Briefly, both CCRMBC56 and *E. coli* S17-1 λpir (pUT::miniTn*5*Cm) from glycerol stocks were initially cultured overnight on LB agar plates at 37 °C. The strains were then subcultured in 10 mL of LB broth overnight at 37 °C in a shaking incubator at 190 rpm. Subsequently, 500 µL aliquots from each liquid culture were combined in a microcentrifuge tube. The mixed bacterial cells were collected by centrifugation, resuspended in 50 µL of LB broth, and then spotted on LB agar plates for conjugation. After incubating the plates at 30 °C overnight, the resulting cultures were resuspended in 1 mL of LB broth. Successful *B. cenocepacia* CCRMBC56 conjugants with miniTn*5*Cm integration into the genome were selected on LB agar plates amended with chloramphenicol (34 µg/mL) and nitrofurantoin (100 µg/mL). About 80,000 colonies of successful conjugants (random miniTn*5*Cm mutants of *B. cenocepacia* CCRMBC56) were pooled, resuspended in 30% glycerol, and stored at −80 °C until the next steps. A 100 µL aliquot from this glycerol stock was inoculated into two sets of 10 mL of LB broth: one with no additives as the control condition and the other supplemented with 1.0 mM CuSO_4_ as the copper-stress condition. Three replicates per condition were prepared, and all cultures were incubated at 28 °C in a shaker at 190 rpm for 24 h. After incubation, bacterial cells from each culture were collected for DNA extraction.

### 2.5. Amplification of MiniTn-Tagged DNA Fragments and DNA Sequencing

Genomic DNA was extracted from the bacterial cells obtained from each culture utilizing GenElute^TM^ Bacterial Genomic DNA Kits following the manufacturer’s guidelines (Sigma-Aldrich, St. Louis, MO, USA). An aliquot of one µg of DNA per sample was utilized for digestion with the restriction enzyme NlaIII (New England Biolabs, Ipswich, MA, USA). The digested genomic DNA samples were precipitated with 0.1 volumes of 3 M sodium acetate and 2 volumes of 95% ethanol overnight at 20 °C and subsequently resuspended in 10 µL of double distilled water. From this, 5 µL was utilized for y-linker adaptor ligation following the protocol developed by Kwon et al. [46]. A polymerase chain reaction (PCR) was performed to select fragments containing miniTn*5* tags, utilizing primers specific to the miniTn*5* transposon and the y-linker adaptor. Amplicons from each sample were sent to the Admera Health Biopharma Services (South Plainfield, NJ, USA) for DNA sequencing. An Illumina NovaSeq S4 (Illumina, San Diego, CA, USA) was used to generate 150 bp long paired-end reads.

### 2.6. Sequencing Analysis of Transposon Insertion Sites

The raw sequence reads were pre-processed with the Trimmomatic tool [47] to eliminate Illumina adapters and filter out reads that had base calls below the quality threshold and lengths shorter than 30 bp. High-quality reads were subsequently refined by eliminating the Y-linker sequence (CTGCTCGAATTCAAGCTTCT; [46]) using the Cutadapt (v 4.9) tool [48], which permitted a maximum error rate of 0.15 (--error-rate: 0.15). The trimmed reads were analyzed using the Bio-Tradis pipeline [49]. In this pipeline, the reads were filtered to include only those with the transposon tag (GGCCAGATCTGATCAAGAGA) for mapping purposes. The transposon tags were removed from these reads, and alignment was conducted against the *B. cenocepacia* reference genome ASM171889v1. The alignment was performed using the Smalt tool with the following specific parameters: mapping quality set to zero to allow for multi-mapping; a k-mer length of 20 and a step size of 1; random assignment of reads that were equally mapped to multiple locations; and a maximum of 10 mismatches permitted in a 100-base read. The alignment output provided plot files for each replicon and library, specifying insertion counts at each nucleotide position. The files were used to compute insertion counts per gene through the tradis_gene_insert_sites built-in function and annotated using the EMBL reference genome file. The generated gene count table was used to predict gene essentiality under each condition using the tradis_essentiality function. The tradis_comparison.R function was utilized to compare fitness effects across conditions to identify genes exhibiting significant differences in mutation rate. Gene set enrichment analysis was conducted utilizing the clusterProfiler tool [50] to identify gene modules essential for bacterial survival in copper-stress environments.

**Figure 1 pathogens-14-00226-f001:**
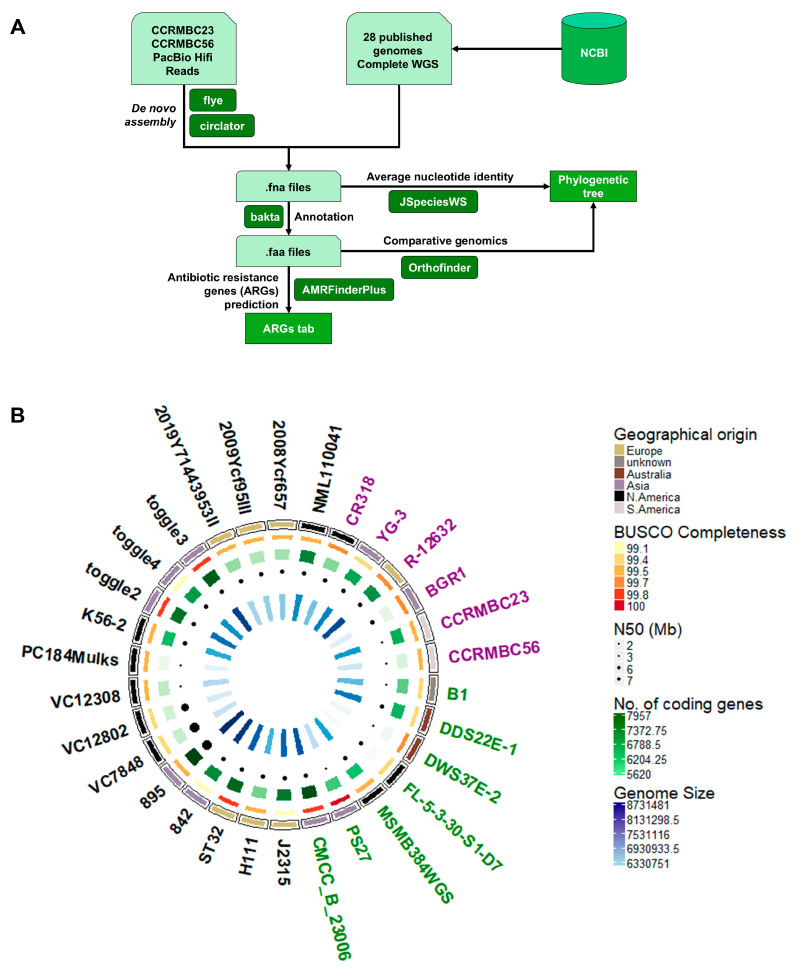
A schematic overview of the pipeline and summary of the comparative genome analysis performed in this study. (**A**) A graphical summary of the methods used to compare the genome assemblies and to predict the antibiotic resistance genes (ARGs) present in each genome. Light green-colored boxes represent the input files used in the analyses; green-colored boxes are the final output files; and the dark green-colored boxes are the tools used for the analyses. (**B**) Genome assemblies used in phylogenomics and prediction of ARGs. The innermost track is the bar plot colored in a blue palette indicating the genome sizes of each assembly. The next track is a dot plot with sizes based on the N50 value of each assembly, assessing the contiguity of each genome. The third track from the middle is a bar plot in a green palette indicating the number of predicted coding genes in each genome. The next track is a heatmap for the completeness of the genome based on the conducted BUSCO analysis using the Burkholderiales database (*n* = 688). The next heatmap track indicates the geographical origin of the *Burkholderia* strains, while the outermost track is the name of the bacterial strain corresponding to each genome assembly. The font color of the strain labels indicates the isolation source of each strain: purple = plants; green = environment; and black = clinical or humans. The figure was created using the circlize package in R (version 4.4.2) [51].

## 3. Results and Discussion

### 3.1. De Novo Assembly of the Genomes of B. cenocepacia CCRMBC56 and B. orbicola CCRMBC23

At the time of the genome sequence analysis for this study, 663 *B. cenocepacia* genome sequences for 641 strains/isolates were deposited in NCBI, of which, only 27 had been at the complete assembly level. One of the important objectives of this study was to generate fully sequenced and annotated genome information for the two Burkholderia strains isolated from diseased onion bulbs. The genomic DNA of *B. cenocepacia* CCRMBC56 and *B. orbicola* CCRMBC23 were sequenced using the PacBio platform, resulting in accurate long reads, enabling us to cover long sequence repeats and to have closed genome assemblies. The de novo assembled genomes were evaluated using the QUAST and BUSCO tools, and the results of this analysis are summarized in Figure 1B and Appendix A. Both strains had four circular contigs, and the genome of *B. cenocepacia* CCRMBC56 was almost 500 kb shorter than that of *B. orbicola* CCRMBC23 (Figure 1B). However, the former had slightly higher GC content than the latter (Appendix A). The length of the largest contigs and the N50 values for *B. cenocepacia* CCRMBC56 and *B. orbicola* CCRMBC23 were 3,639,184 bp and 2.6 Mb, and 3,691,612 bp and 2.9 Mb, respectively. Despite these slight differences in structural features of these genomes, it was found that they had the same completeness level (99.7) based on the BUSCO assessment (Figure 1B; Appendix A).

The gene annotation analysis revealed that the number of protein-coding sequences (CDS) found in both strains differed based on their genome sizes. However, the coding density for CCRMBC56 was 87.8, which was 0.3 higher than that of strain CCRMBC23. All other information on genetic features is summarized in Appendix A. As we were able to assign functions to 97.12% and 95.62% of the genes for *B. cenocepacia* CCRMBC56 and *B. orbicola* CCRMBC23, respectively, the conducted gene calling analysis was overall successful.

### 3.2. Comparative Genomic Analysis

For comparative genomic analysis, the de novo assembled genomes of strains *B. cenocepacia* CCRMBC56 and *B. orbicola* CCRMBC23 were compared to previously assembled *B. cenocepacia* genomes published in the NCBI database (Figure 1B and Appendix A) and to the B. glumae BGR1 genome (the outgroup). The 30 genomes originated from five continents: Asia, Australia, Europe, and North and South America. Most of the strains (57%) were isolated from humans, and 23% and 20% of the strains were collected from the environment (aerosol, soil, etc.) and plants, respectively. Pairwise ANIs for all strains were computed to estimate the genetic relatedness of the bacterial strains [52], which resulted in three clades excluding the outgroup control, B. glumae BGR1 (Figure 2 and Appendix A).

The strains DDS22E-1 and DWS37E-1 from clade 1, isolated from aerosol and soil samples, were first identified as *B. cenocepacia* strains [53]. Nonetheless, their ANI values, in comparison to other *B. cenocepacia* stricto sensu strains from cluster 2, varied between 88.58% and 89.17% (Figure 2). This suggests that these two strains should be considered as distinct species from *B. cenocepacia*, as they demonstrate considerable genetic diversity and taxonomic distinction [54,55]. Moreover, while these two strains are classified within one clade, their ANI values fall below 90%, implying intra-specific genetic relationships. The second clade harbored 16 strains of *B. cenocepacia*, 87.5% of which were derived from human samples; hence, we classified this as a human-pathogenic clade in this study. Notably, *B. cenocepacia* CCRMBC56 was clustered into this clade, indicating that this strain, isolated from a diseased onion sample, may survive and cause complications in human hosts due to its high genetic similarity, approximately 98%, with isolates obtained from clinical patients. Finally, clade 3, including *B. orbicola* CCRMBC23, consisted of strains derived from various isolation sources and geographical locations and was classified as a mixed clade in this study. The ANI values of the strains inside this clade varied from 93.76% to 94.82% compared to those in the human pathogenic clade, indicating intermediate relatedness to the human pathogenic strains [56]. On the other hand, the ANI values of the strains inside the mixed clade, excluding strains R-12632 and YG-3, varied from 97.41% to 99.15%, indicating that these strains must be classified under the same Burkholderia species, separate from *B. cenocepacia*. In a recent systematic study on Burkholderia species [24], it was observed in the phylogenetics analyses that the strains FL-5-3-30-S1-D7, VC12802, CR318, VC7848, and PC184Mulks formed a sub-clade together with the *B. orbicola* strain TAtl-371T, although it is classified as *B. cenocepacia* in GenBank. In addition, it is common to find strains misclassified as *B. cenocepacia* that are positioned in the vicinity of *B. cenocepacia* and *B. orbicola* based on phylogenetic analyses using the core genome [24,57,58]. The remaining strains in this clade—R-12632 and YG-3—must belong to species other than *B. cenocepacia* and *B. orbicola* based on their ANI values when compared to other strains. Overall, the results of our ANI analyses imply the need for a reclassification of some *B. cenocepacia* strains whose genomes are deposited in GenBank.

**Figure 2 pathogens-14-00226-f002:**
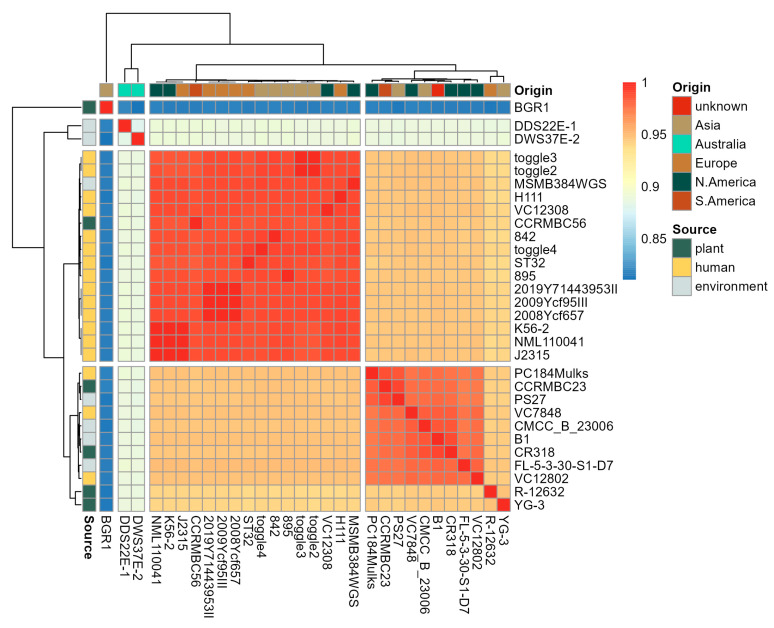
Hierarchical cluster that demonstrates the genetic relatedness of all *Burkholderia* genomes compared in this study. Average nucleotide identity (ANI) values computed using JSpeciesWS tool [42] were used for computing similarities and clustering bacterial genomes. Distances were visualized using the pheatmap package in R [59]. The leftmost and the topmost sidebars represent the isolation source and the geographical origin of the bacterial strains. The heatmap color spectrum from blue to red represents the minimum (0.810407) to maximum (1.0) ANI values, respectively.

### 3.3. ARG Prediction

One identified approach to mitigate the further spread of AMR is conducting ARG surveillance, where algorithms for prediction can be useful in tracking the development of antibiotic resistance across diverse environments and ecosystems [60,61,62]. Here, we utilized genomic information to predict the different ARGs across other Burkholderia species downloaded from the NCBI database. Overall, we detected four genes for aminoglycosides resistance; one gene for arsenic resistance; five genes for β-lactam resistance; six genes for mercury toxicity resistance; two genes for phenicol resistance; one gene for rifamycin resistance; and two genes for tetracycline resistance (Figure 3). Generally, most strains in this analysis possess genes for resistance to β-lactam, rifamycin, and tetracycline. The beta-lactamase gene variants bla, blaOxa, and blaPen-B were found present in 100%, 93.3%, and 90% of the strains, respectively. Fragments of the ampC gene, with 52–54% identity, were also found in 28 strains. Strain toggle4 is the only isolate with five resistance genes to β-lactam, while the other 80% of the strains have four. The presence of multiple versions and types of β-lactam resistance genes in a bacterial species enables them to resist a broader spectrum of β-lactam antibiotics [63]; to become more robust against β-lactamase inhibitors [64]; and to survive in diverse environments and under different antibiotic pressure [65].

Beta-lactam antibiotics have efficacy against a broad spectrum of bacteria, including several Gram-positive, Gram-negative, and anaerobic species. This extensive range of activity renders them effective for addressing diverse illnesses caused by bacterial infections with little harm to human cells [67]. However, the rising incidence of resistance mechanisms and possible adverse effects complicate their use. Current research and development initiatives aim to address these limitations while maintaining the advantages of this crucial class of antibiotics. The re-potentiation of β-lactam antibiotics through the utilization of copper oxide nanoparticles represents a novel strategy to address antibiotic resistance and improve the effectiveness of these critical medications. Arul Selvaraj et al. [68] reported that the combination of β-lactam antibiotics, particularly amoxiclav (amoxicillin/clavulanic acid), with biogenic copper oxide nanocubes can enhance their efficacy against multidrug-resistant bacteria. The re-potentiation effect is likely mediated by several mechanisms: (1) copper oxide nanoparticles may facilitate the entry of β-lactams into bacterial cells [68,69]; (2) these nanoparticles could disrupt bacterial resistance mechanisms, including efflux pump activity [70]; and (3) copper ions released from the nanoparticles may exert direct antimicrobial effects, thereby enhancing the efficacy of β-lactams [71]. This approach illustrates the capacity of nanomaterials to enhance current antibiotics, presenting a viable strategy to combat the increasing issue of antibiotic resistance. This report seeks to elucidate the copper resistance of the onion-pathogenic strains of *B. cenocepacia*, which will aid in the development of more effective strategies for re-potentiating β-lactam antibiotics, potentially surmounting current resistance mechanisms and establishing novel combination therapies to address antibiotic-resistant infections.

### 3.4. Identification of Essential Genes for Normal and Copper-Stressed Conditions

To identify essential genes, transposon insertion events were quantified first for each gene: the sequence reads with transposon tags were filtered for the downstream analyses. Overall, the number of reads tagged with a transposon was only around 18% of the sequence data obtained, of which, 47% were mapped to the reference genome (Table 1). Transposon saturation was computed by dividing the total sequence length by the number of unique insertion sites. The results indicated that transposon insertions were distributed over the whole genome, with an average insertion of one transposon for every 91 base pairs (Table 1).

After quantifying the transposon insertion events across the whole genome, each gene was scored with an insertion index, which was computed by dividing the total number of transposon insertions per gene by its respective gene length. Then, genes were called essential when they had an insertion index lower than the computed cutoff for each condition. Under normal conditions, the computed value for an ambiguous changepoint was 0.0148 (Figure 4A), which means that genes with an insertion index higher than this value were called non-essential genes for the bacteria under normal growing conditions. The computed essential changepoint value for the same condition was 0.0101 and genes with insertion indices lower than this value were called essential. Under the copper-stressed conditions, the computed changepoint values for essential and ambiguous were 0.0099 and 0.0143, respectively (Figure 4B), and 357 genes were called essential due to having insertion indices lower than 0.0099. In summary, the number of essential genes identified from these two conditions was almost the same.

The gene essentiality data were subjected to gene set enrichment analysis to identify the genes or gene modules that are responsible for the survival of the strain CCRMBC56 under each condition. The results showed that the genes that were rarely found to have mutations under normal conditions are the housekeeping genes that are important for bacterial growth and multiplication (Figure 5A). On the other hand, aside from these housekeeping genes, the genes for DNA integration and recombination, and cation transport were found to be essential under copper-stressed conditions (Figure 5B).

In our results, we demonstrated that the genes involved in cellular growth and multiplication are essential for the survival of the bacteria under normal growth conditions. However, aside from these genes, those involved in DNA integration and recombination, and in cation transport were also found to be essential for the survival of the bacteria upon exposure to copper stress. Copper, although essential in minute quantities, exhibits toxicity at elevated concentrations. The toxicity of copper towards bacterial organisms is a complex phenomenon that encompasses a multitude of mechanisms, such as inhibition of lipoprotein maturation, resulting in the accumulation of deleterious precursors within intracellular compartments [73]; damage to the cell envelope causing cell lysis [74]; and production of reactive oxygen species (ROS), leading to oxidative stress and cell death [21]. Cation transport systems are of utmost importance in bacterial adaptation to copper stress as they control the import and export of copper ions, maintain intracellular equilibrium, and avert toxicity via effective export mechanisms. The implementation of these adaptive strategies is critical for the survival of microorganisms in environments characterized by fluctuating copper concentrations [75].

The process of DNA integration and recombination is crucial for bacterial adaptation to copper stress [76]. It helps maintain genetic integrity and enables the development of stress response systems. The expression of DNA repair genes is altered in response to copper stress. Tripathi et al. [77] reported that exposure to copper may lead to the activation and deactivation of some DNA repair genes. In their study, resolvase encoded by ruvC, which plays a role in eliminating Holliday junctions during DNA repair, was found to be overexpressed in the presence of copper stress. Their report was the first to suggest that the removal of Holliday junctions is crucial for the survival of Desulfovibrio alskensis under copper-stressed conditions. This mechanism is crucial for preserving genetic integrity. Moreover, oxidative stress caused by copper may result in damage to cellular components, including DNA [20,78,79]. This emphasizes the need for effective DNA repair pathways in preserving genomic integrity [80].

### 3.5. Identification of Beneficial and Deleterious Gene Mutations for Resistance to Copper Stress of B. cenocepacia CCRMBC56

To determine whether the mutations in a gene were deleterious or beneficial for the strain CCRMBC56 under copper-stress conditions, a fitness effect analysis was conducted. Through this analysis, we identified differentially present genes under copper-stress conditions in comparison with normal growth conditions (Appendix A and Appendix A). The underrepresented genes are the genes that were rarely found to have transposon insertions, indicating that these genes, when mutated, would be deleterious for the bacteria under copper-stress conditions. Overrepresented genes, on the other hand, were found to have more than the expected number of transposon insertions in them, suggesting that mutation in these genes would be beneficial for bacterial survival under copper-stress conditions.

Gene set enrichment analysis was conducted on the differentially present genes (Appendix A) to determine the modular functions of these genes (Figure 6). The results from this analysis suggested that mutations in genes related to the oxidative stress response, cation transport, ABC transporter, and the cell outer membrane are deleterious for the bacteria under copper-stressed conditions. On the other hand, mutations in genes involved in the TCA cycle, penicillin binding, and FAD binding are likely beneficial for the survival of this bacteria under copper-stress conditions.

The fitness effect analysis revealed that the mutation of genes involved in the oxidative stress response, cation transport, ABC transporter, and the cell outer membrane is deleterious for the organism under copper stress. Bacteria exhibit an upregulation of oxidative stress resistance genes in response to copper-induced stress [73,78]. In Staphylococcus aureus, the presence of genes responsible for the oxidative stress response is essential for its proliferation in environments with elevated copper concentrations. Mutants defective in important genes related to oxidative stress, such as ahpC, katA, sodA, and sodM, exhibited heightened sensitivity to copper [78,81]. This emphasizes the significance of superoxide dismutase functions in reducing the harmful effects of copper [78,82]. Additionally, bacteria have developed many strategies to combat the harmful effects of copper, such as the use of multi-copper oxidases, which serve as a defense mechanism against copper toxicity [20,83]. Under conditions of excessive copper concentrations, the activation of bacterial copper tolerance genes is often facilitated by specific transcriptional regulators, such as CueR and GolS [84,85]. These regulators control the expression of genes that are responsible for copper tolerance and the response to oxidative stress [20].

Bacterial copper homeostasis relies on copper transport systems, such as ABC transporters. These mechanisms aid in controlling the concentration of copper ions inside cells, limiting harmful levels of toxicity by exporting cytoplasmic copper to the periplasm while maintaining a sufficient amount for vital cellular functions. The ABC transporter is likely responsible for maintaining this equilibrium by facilitating the entry of copper ions into the cell [75,86].

On the other hand, the fitness effect analysis also showed that mutations in genes involved in the TCA cycle, penicillin binding, and FAD binding are beneficial for the survival of the bacterial strain under copper-stressed conditions. The TCA cycle is a significant metabolic pathway responsible for producing reducing equivalents, such as FADH2 and NADH, which are vital for the operation of the electron transport chain and the synthesis of ATP. Nevertheless, the TCA cycle may also generate ROS, especially via complex II, during regular oxygen consumption. During cellular energy production, this mechanism might result in the formation of ROS as byproducts [87]. Within the context of copper stress, which triggers oxidative stress, the involvement of the TCA cycle in the generation of ROS might intensify the damage to cells and affect the ability of bacteria to survive in high-copper environments [78]. Inactivation of this ROS-generating pathway through enforcing mutations in genes involved in the TCA cycle could aid the bacterial cells in preventing the deleterious effects of intracellular ROS imbalances due to copper stress.

## 4. Conclusions

This study used genome-sequence-based approaches to identify antibiotic resistance genes in onion-pathogenic *Burkholderia* strains. Complete whole-genome sequences of the *B. cenocepacia* strain CCRMBC56 and *B. orbicola* strain CCRMBC23 were characterized and used to predict the presence of ARGs. The findings from our ARG prediction analysis may provide additional information for the surveillance of the ARGs in the environment. This research also illustrates the application of a large-scale transposon-mediated mutant library combined with next-generation sequencing to identify genes critical for the survival of *B. cenocepacia* strain CCRMBC56 under copper stress. This approach serves as an effective method for assessing gene essentiality in bacterial species. The gene essentiality analysis indicated that, alongside genes associated with cell growth and division, genes responsible for DNA integration, DNA recombination, and cation transport are critical for the survival of the *B. cenocepacia* strain CCRMBC56 under toxic copper exposure. The fitness analysis indicated that mutations in genes associated with the oxidative stress response, cation transport, ABC transport, and the outer cell membrane are disadvantageous for the survival of *B. cenocepacia* CCRMBC56 under copper stress, while mutations in genes related to the TCA cycle, penicillin binding, and FAD binding are advantageous. Although additional laboratory tests are required to validate the predicted role of these gene in copper resistance, this study provides valuable insights into potential targets to reduce the resistance of *Burkholderia* spp. to antibiotic compounds including copper products.

## Figures and Tables

**Figure 3 pathogens-14-00226-f003:**
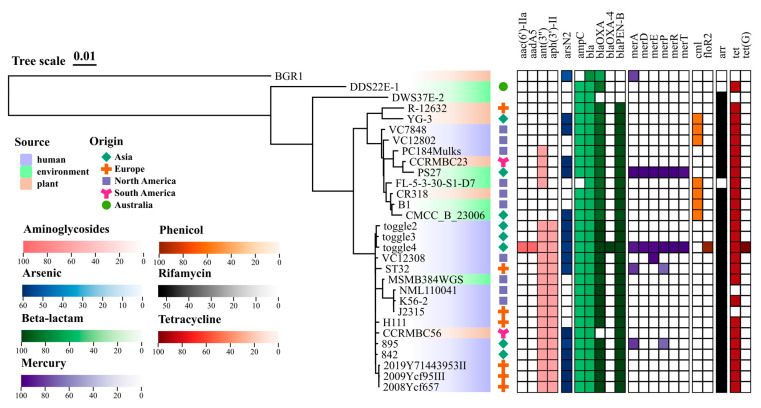
Genome sequence-based antibiotic resistance gene (ARG) prediction across the strains of *Burkholderia glumae*, *B. cenocepacia*, *B. orbicola*, and closely related strains of *Burkhoderia* sp. The phylogenetic tree was created based on the core orthogroups found across all strains [44]. The heatmap represents the percent identity of the ARGs found in each strain using the NCBI AMRFinderPlus tool [45]. Data were combined and visualized using the online tvBOT tool [66].

**Figure 4 pathogens-14-00226-f004:**
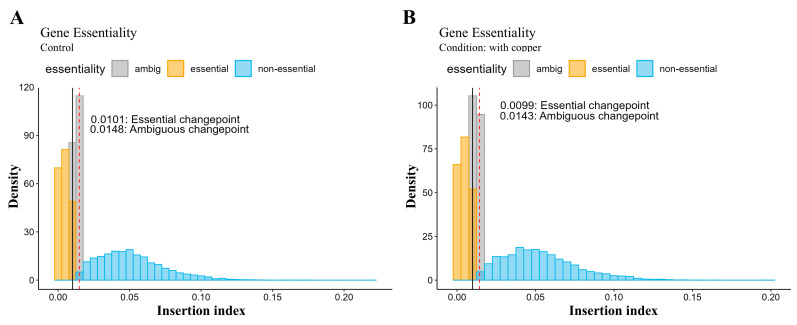
Density plots showing the distribution of genes per insertion index score for *Burkholderia cenocepacia* CCRMBC56 grown under normal (**A**) and copper-stressed (**B**) conditions. Yellow, gray, and blue bars indicate the essential, ambiguous, and non-essential genes, respectively. The bold black vertical line specifies the essential changepoint cut-off while the broken red vertical line signifies the ambiguous changepoint cut-off. Data obtained from the gene essentiality analysis [49] were plotted into histograms using the ggplot2 package [72] in R.

**Figure 5 pathogens-14-00226-f005:**
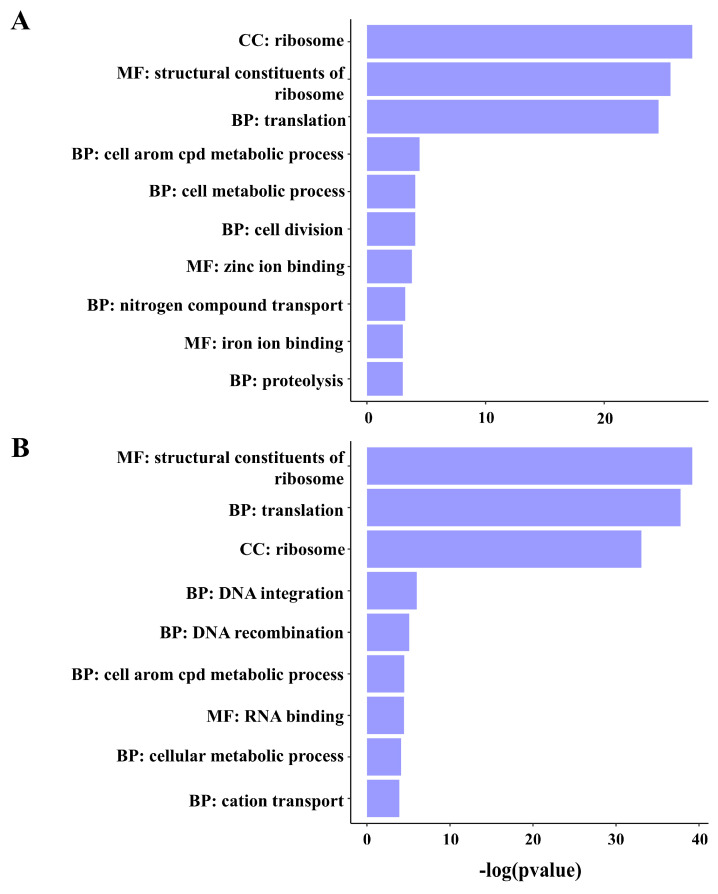
Gene set enrichment analysis reveals the functions of the genes identified to be essential for *Burkholderia cenocepacia* CCRMBC56 to survive under normal (**A**) and copper-stressed (**B**) conditions. The data obtained from the gene essentiality analysis [49] for each condition were used for gene enrichment analysis using the clusterProfiler package in R [50].

**Figure 6 pathogens-14-00226-f006:**
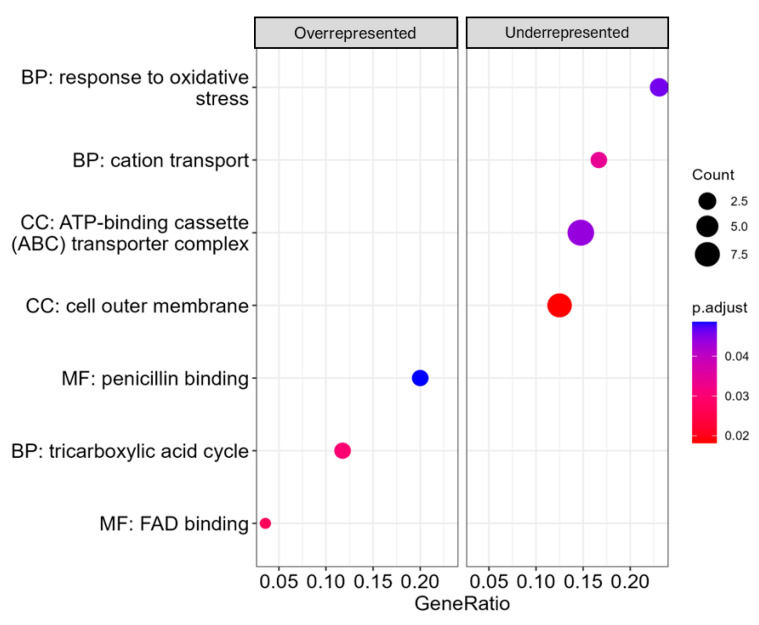
Fitness contributions of mutated genes. Under copper-stress conditions, mutations in overrepresented genes are considered beneficial for the survival of *Burkholderia cenocepacia* CCRMBC56, while the mutations in underrepresented genes are deleterious. Log-fold change data obtained from the fitness contribution analysis [49] were used for the gene set enrichment analysis using the clusterProfiler package in R [50].

**Table 1 pathogens-14-00226-t001:** Mapping statistics of the transposon-tagged DNA fragments isolated from strain CCRMBC56 grown under normal and copper-stressed conditions.

Sample ^1^	Total Reads	Reads Matched	% Matched	Reads Mapped	% Mapped	Total Unique Insertion Sites	Total Seq Len/Total UIS
Control 1	15,089,775	2,487,624	16.48549	1,109,433	44.5981	88,092	88.32355
Control 2	9,778,537	1,862,460	19.04641	889,442	47.7563	78,646	98.9319
Control 3	15,373,016	2,460,912	16.008	1,138,419	46.26005	85,234	91.28514
Condition 1	9,686,912	1,801,336	18.59556	833,144	46.25145	76,117	102.2189
Condition 2	12,189,030	2,360,912	19.36915	1,140,352	48.30133	84,853	91.69503
Condition 3	13,791,454	2,605,272	18.89048	1,322,081	50.74637	100,160	77.68169

^1^ Control—samples extracted from cultures grown in normal LB medium; Condition—samples collected from cultures grown in LB medium supplemented with 1.0 mM CuSO_4_. Presented data were obtained from the analysis conducted using the Bio-TraDIS pipeline [49].

## Data Availability

The data supporting the results of this study can be obtained from the corresponding author upon reasonable request. The raw DNA sequence data for *Burkholderia cenocepacia* CCRMBC56 and *B. orbicola* CCRMBC23 were deposited in the National Center for Biotechnology Information (NCBI) database under the BioProject ID PRJNA1074852.

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
