# Peer review of "Characterization of the Antibiotic and Copper Resistance of Emergent Species of Onion-Pathogenic Burkholderia Through Genome Sequence Analysis and High-Throughput Sequencing of Differentially Enriched Random Transposon Mutants"

_pathogens, 2025, doi:10.3390/pathogens14030226_

Round 1
Reviewer 1 Report
Comments and Suggestions for Authors
The manuscript titled “Characterization of the copper-resistance of emergent species of onion-pathogenic Burkholderia through genome sequence analysis and high-throughput sequencing of differentially enriched random transposon mutants” is devoted to analysis of antimicrobial resistance (AMR) in two emergent onion-pathogenic species of Burkholderia, B. cenocepacia CCRMBC56 and B. orbicola CCRMBC23. The genes contributing to the copper resistance of B. cenocepacia CCRMBC56 were identified through high-throughput analysis of mutated genes in random transposon mutant populations differentially enriched in copper-containing medium. Results indicated that genes involved in DNA integration, recombination, and cation transport are important for the survival of B. cenocepacia CCRMBC56 in copper-stressed conditions. Furthermore, the fitness effect analysis identified additional genes crucial for copper resistance, which are involved in the functions associated with oxidative stress response, the ABC transporter complex, and the cell outer membrane.
The manuscript is well-written and reports reliable and valuable data. The only problem – the title and abstract missed description of the first valuable part of this work – analysis of antibiotics resistance genes in the analyzed genomes.
It looks like the Authors originally devoted more attention to antibiotic resistance, but for some reason decided to concentrate on copper resistance. Some parts of the introduction still talk about antibiotics, like at the Lines 31-36 “The advent of antibiotics has significantly reduced post-surgical infections, leading to a marked decline in mortality rates associated with bacterial infections [1, 2]. Additionally, the development of antibiotics has extended human lifespan throughout the 20th century [3]. Nevertheless, intensive usage of antibiotics creates substantial selection pressure, facilitating the survival and reproduction of antibiotic-resistant bacteria, thereby significantly contributing to the spreading of antibiotic-resistant microorganisms [4-6].
And at Line 37: “Antimicrobial resistance (AMR) occurs when pathogenic microbes undergo genetic changes that confer resistance to antibiotic treatments designed to eliminate them.”
And at the Line 307 “Generally, most strains in this analysis possess genes for resistance to β-lactam, rifamycin, and tetracycline.”
Please, modify the title and abstract to reflect the content devoted to antibiotic resistance too.
Author Response
Comment 1:
The manuscript is well-written and reports reliable and valuable data. The only problem – the title and abstract missed description of the first valuable part of this work – analysis of antibiotics resistance genes in the analyzed genomes.
It looks like the Authors originally devoted more attention to antibiotic resistance, but for some reason decided to concentrate on copper resistance. Some parts of the introduction still talk about antibiotics, like at the Lines 31-36 “The advent of antibiotics has significantly reduced post-surgical infections, leading to a marked decline in mortality rates associated with bacterial infections [1, 2]. Additionally, the development of antibiotics has extended human lifespan throughout the 20th century [3]. Nevertheless, intensive usage of antibiotics creates substantial selection pressure, facilitating the survival and reproduction of antibiotic-resistant bacteria, thereby significantly contributing to the spreading of antibiotic-resistant microorganisms [4-6].
And at Line 37: “Antimicrobial resistance (AMR) occurs when pathogenic microbes undergo genetic changes that confer resistance to antibiotic treatments designed to eliminate them.”
And at the Line 307 “Generally, most strains in this analysis possess genes for resistance to β-lactam, rifamycin, and tetracycline.”
Please, modify the title and abstract to reflect the content devoted to antibiotic resistance too.
Response:
We appreciate the positive comments on our manuscript. We also thank you for the valuable suggestion about the title and abstract. We did not separate antibiotic resistance and copper resistance strictly because we considered copper as an antibiotic compound. However, your comments indicate that some confusion may arouse because of this vagueness.
Complying with your suggestion, we have changed the title to ‘Characterization of the antibiotic and copper resistance of emergent species of onion-pathogenic Burkholderia through genome sequence analysis and high-throughput sequencing of differentially enriched random transposon mutants.’
In the abstract, ‘their high level of copper resistance’ has been changed to ‘their high levels of antibiotic and copper resistance’ (Line 15-16).
Reviewer 2 Report
Comments and Suggestions for Authors
In this manuscript, the authors made important contributions to understanding of genes potentially associated with their high level of copper resistance in two emergent onion pathogenic species of Burkholderia, B. cenocepacia CCRMBC56 and B. orbicola CCRMBC23. The results shows the genome analysis for both bacterial strain compared with related bacterial species and antibiotic resistance genes. Also, the authors made transposome library by conjugating B. cenocepacia CCRMBC56 with E. coli S17-1λpir (pUT::miniTn5Cm). The results analysis were made under control condition and the other supplemented with 1.0 mM CuSOâ‚„ as the copper-stress condition. Essential genes were analyzed and reported. The research work is very interesting, all results are well-performed and clearly presented.
Author Response
Comment 1:
In this manuscript, the authors made important contributions to understanding of genes potentially associated with their high level of copper resistance in two emergent onion pathogenic species of Burkholderia, B. cenocepacia CCRMBC56 and B. orbicola CCRMBC23. The results shows the genome analysis for both bacterial strain compared with related bacterial species and antibiotic resistance genes. Also, the authors made transposome library by conjugating B. cenocepacia CCRMBC56 with E. coli S17-1λpir (pUT::miniTn5Cm). The results analysis were made under control condition and the other supplemented with 1.0 mM CuSOâ‚„ as the copper-stress condition. Essential genes were analyzed and reported. The research work is very interesting, all results are well-performed and clearly presented.
Response:
We greatly appreciate the positive comments on our manuscript!